# Rationale for the shielding policy for clinically vulnerable people in the UK during the COVID-19 pandemic: a qualitative study

Alison Porter ,[1] Ashley Akbari ,[1] Andrew Carson-Stevens,[2] Jeremy Dale ,[3] Lucy Dixon,[4] Adrian Edwards ,[2] Bridie Evans,[1] Lesley Griffiths,[4] Ann John ,[1] Stephen Jolles,[2] Mark Rhys Kingston,[1] Ronan Lyons ,[1] Jennifer Morgan,[5] Bernadette Sewell,[6] Anthony Whiffen,[7] Victoria Angharad Williams,[1] Helen Snooks[1]

For numbered affiliations see end of article.

**Correspondence to**
Dr Alison Porter;
A.M.Porter@swansea.ac.uk

## ABSTRACT

**Introduction** Shielding aimed to protect those predicted to be at highest risk from COVID-19 and was uniquely implemented in the UK during the first year of the pandemic from March 2020. As the first stage in the EVITE Immunity evaluation (Effects of shielding for vulnerable people during COVID-19 pandemic on health outcomes, costs and immunity, including those with cancer:quasi-experimental evaluation), we generated a logic model to describe the programme theory underlying the shielding intervention.

**Design and participants** We reviewed published documentation on shielding to develop an initial draft of the logic model. We then discussed this draft during interviews with 13 key stakeholders involved in putting shielding into effect in Wales and England. Interviews were recorded, transcribed and analysed thematically to inform a final draft of the logic model.

**Results** The shielding intervention was a complex one, introduced at pace by multiple agencies working together. We identified three core components: agreement on clinical criteria; development of the list of people appropriate for shielding; and communication of shielding advice. In addition, there was a support programme, available as required to shielding people, including food parcels, financial support and social support. The predicted mechanism of change was that people would isolate themselves and so avoid infection, with the primary intended outcome being reduction in mortality in the shielding group. Unintended impacts included negative impact on mental and physical health and well-being. Details of the intervention varied slightly across the home nations of the UK and were subject to minor revisions during the time the intervention was in place.

**Conclusions** Shielding was a largely untested strategy, aiming to mitigate risk by placing a responsibility on individuals to protect themselves. The model of its rationale, components and outcomes (intended and unintended) will inform evaluation of the impact of shielding and help us to understand its effect and limitations.

## STRENGTHS AND LIMITATIONS OF THIS STUDY

⇒ This paper presents the first description of the rationale for shielding which was an internationally unique and untested public health intervention implemented in the UK during the COVID-19 pandemic.
⇒ Our paper combines formal documentation on the shielding programme in the UK with interviews with those involved in creating and implementing it, so allowing for an exploration of how the rapidly implemented policy was operationalised on the ground.
⇒ This logic model provides the first step in undertaking the EVITE Immunity study (Effects of shielding for vulnerable people during COVID-19 pandemic on health outcomes, costs and immunity, including those with cancer:quasi-experimental evaluation), a population-scale national assessment of effects of shielding on COVID-19 infection rate, mortality, serious illness, use of National Health Service resources, health-related quality of life and behaviour.
⇒ While we collected views from policy makers in England and Wales, the majority of interview participants were based in Wales, so their experience may not be representative of all other parts of the UK.
⇒ Developing this logic model within EVITE Immunity study has involved people with direct experience of shielding from the outset, with public contributors represented across all aspects of research development and implementation, reflecting strong views that evidence about effects of shielding is needed.

## BACKGROUND

As an early response to the COVID-19 pandemic, the four UK nations introduced a policy of 'shielding' for clinically extremely vulnerable (CEV) people. Those identified as being at the highest risk from COVID-19 infection, due to pre-existing conditions such as lung disease or current immunosuppressant medications, were strongly advised to strictly self-isolate, not leaving the home unless it was vital. The policy was the subject of rapid development and implementation. It was first discussed by the UK's Scientific Advisory Group for Emergencies (SAGE) on 13 March 2020 and put into place within 10 days.

To support the shielding policy, a programme of practical and financial support was made available by a range of statutory, commercial and third sector partners.

The shielding intervention was in place for a total of 10 months over two periods, before being suspended in the spring of 2021. It eventually included over 4 million people across the UK.[1]

Shielding was introduced as a novel precautionary response to an unprecedented situation, with no underpinning empirical evidence about its effectiveness at reducing SARS-CoV-2 infections, serious illness or deaths.

We undertook an evaluation of shielding in Wales (EVITE Immunity—Effects of shielding for vulnerable people during COVID-19 pandemic on health outcomes, costs and immunity, including those with cancer:quasi-experimental evaluation), where records for the 130 000 people who were identified for shielding are already anonymously linked with other integrated data sources, using the Medical Research Council (MRC)-funded ConCOV (*Controlling COVID19 through enhanced population surveillance and intervention*) project in the Secure Anonymised Information Linkage Databank.[2] Initial findings show that people were more likely to have been identified for inclusion in the shielding intervention with increasing age, frailty and residence in deprived areas; and that reported infection rate was higher in the shielded cohort than the non-shielded general population, though testing rates were higher and infection rates among those not tested in each cohort were unknown.[3] We will also report how shielding affected deaths, healthcare utilisation, immunity status, National Health Service (NHS) costs and quality of life, and how people complied with guidance.[4]

In line with the latest MRC guidelines on evaluating complex interventions,[5] the first stage of the EVITE Immunity study aimed to develop a programme theory to explain the intentions of the policy, making explicit all components of the intervention (defined here as being the shielding policy plus support programme), and representing these in a logic model, presented in this paper. Logic models can have a particular value in helping to articulate causality in the evaluation of public health interventions.[6] We will use this logic model to guide the analysis and interpretation of evaluation findings.

## METHODS

Our study was designed as case study research. Based on published information, we prepared a draft logic model describing the components of the intervention, the mechanisms by which it was assumed to work, outcomes, intended impact, risks and relevant contextual factors. We conducted individual interviews (n=3) and group interviews (n=4) online with a total of 13 key stakeholders: senior policy makers and clinicians from public health and chief medical officers' teams, and representatives from local government and the voluntary and community sector (VCS), in Wales (n=12) and England (n=1). Respondents were recruited to provide a range of relevant perspectives. Interviews were conducted by experienced qualitative researchers from Swansea University. In advance of the interview, we shared an information sheet on the study with participants and they completed written consent forms.

We used a semistructured interview schedule (online supplemental appendix) to explore the rationale for shielding, steps undertaken to create and implement the intervention and individual/organisational roles. We showed participants the draft logic model and invited comments and discussion. Data collection took place between March and May 2021.

We recorded and transcribed interviews, with participants' consent. We analysed the data using a framework approach to thematic analysis, incorporating a tabular data summary of cases/codes and data extracts,[7] and refined the logic model into a final version. Each transcript was reviewed and coded by two members of the study team; findings and implications were discussed by the whole study team, including public participants.

### Patient and public involvement

People affected by the shielding policy have been directly involved throughout development of the research design.[8–10] Two were coapplicants on the funding proposal and are members of the Research Management Group overseeing study implementation (LG, LD). They work with six more public contributors via a patient advisory panel. An independent study steering committee includes two further public contributors. Our public contributors and some academic coapplicants were personally, directly or indirectly affected by the implementation of the shielding policy.

## RESULTS

The final version of the logic model describing the shielding intervention is shown in figure 1. This incorporates some changes made following analysis of the interviews, including reduced deaths among the shielding population being highlighted as the primary intended impact, while the proximate outcome of 'avoided infection' was changed to 'reduced infection'. Small additions were made to inputs, context and unintended impacts. Below, we present and discuss the key aspects of the logic model and report the experience and reflections of the stakeholders we interviewed.

### Inputs: components of the intervention

The intervention was complex. There were three core processes relevant to all people advised to shield: the chief medical officers of the four nations reaching agreement on the clinical criteria for inclusion on the CEV list; identification of people to be added to the list; and communication with those identified.

Identifying and communicating with CEV people took place in phases, with batches of individuals being added to the list over the time the shielding programme was

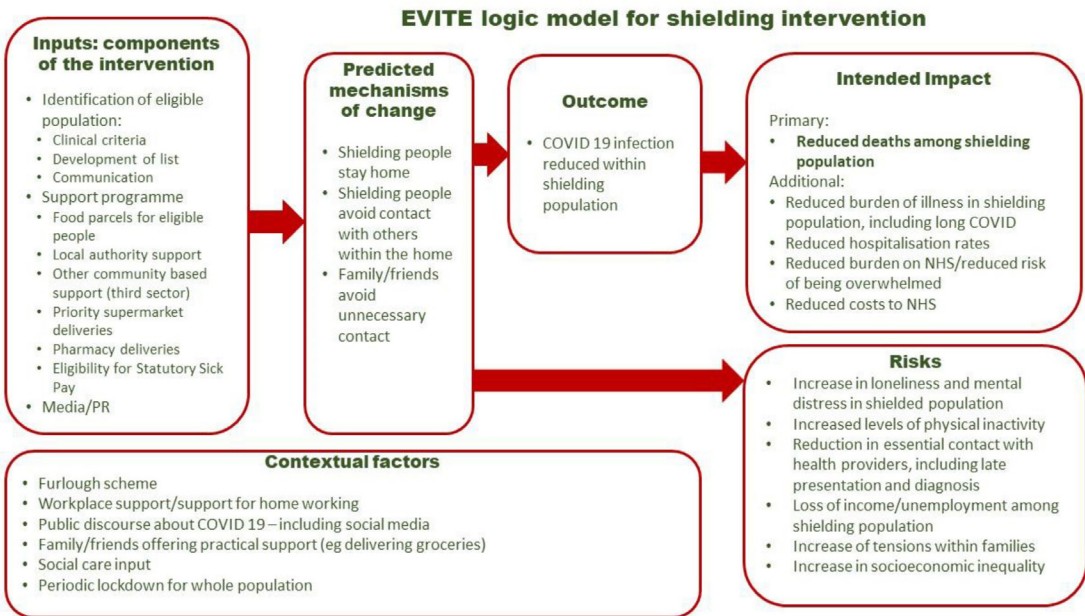

**Figure 1** Logic model describing the shielding intervention. NHS, National Health Service; PR, public relations.

'live' and much smaller numbers being removed. Across the UK, CEV people were identified through searches of centralised databases (which produced the majority of names for the list in Wales), and through primary and secondary care records, in conjunction with doctors' clinical judgement.[11] This mixed approach was described as '*build the list nationally first and then ask GPs [general practitioners] to review*' (Participant 2, policy maker). Some GPs and practice managers responded to people requesting to be removed from, or added to, the list. As one respondent noted, this was more significant than a mere administrative process:

> The GP had the authority to put people on and off, but a lot of the GPs hadn't realised… what the implication of that was for the individual about going back to work, back to school, or actually getting access to the food box or not. (Participant 9, local government)

The development of the QCovid risk prediction tool, which identified clinical and demographic risk factors for COVID-19-related hospitalisations and deaths,[12 13] led to substantial numbers being added to CEV lists in the summer of 2020.

The 'binary' nature of inclusion on the list was seen by one respondent as problematic:

> It didn't reflect the spectrum of risk. So you were selecting a group of people for really quite an austere set of advisory statements, but actually nothing for the rest. (Participant 1, policy maker)

Letters were sent to CEV people strongly advising them to shield from 23 March 2020, initially for 12 weeks.[14] Respondents reported considerable thought going into the exact wording of the letters, with an emphasis on them giving advice, not instruction:

> We were telling people they could rather than telling people they must. ….It was strong advice, but it was set in advisory terms. (Participant 7, policy maker)

Where telephone numbers were available, reinforcement of the advice was sent by text message.

The other aspect of the intervention was the support programme, a range of components experienced by some CEV people as required. These included eligibility (achieved through a change in legislation) for statutory sick pay for CEV people, even if they were not ill, to allow them to take time off work if their job could not be done at home. For those who needed them, the UK government contracted with commercial providers to deliver weekly food parcels. CEV people could also access delivery of medications and were given priority by supermarkets for food delivery.

Other forms of support were available through local government and partner organisations, and CEV people were invited to register to access this. Welsh local government respondents described taking a proactive approach to contacting CEV people who might be in need of help through welfare telephone calls:

> Trying to keep [CEV people] linked into their society, their communities. So it was very much a kind of social response to it…Most of the people doing the calling were our librarians. (Participant 10, local government)

Although changes to data sharing rules allowed lists of CEV people to be supplied to local government, significant challenges were reported with using the shared data:

> It was incomplete. It wasn't accessible…It wasn't transferable. It didn't match any of the other datasets. We couldn't identify individuals through it. I

think the first lot didn't even have people's names. It just had their NHS number. (Participant 11, local government)

VCS organisations often worked in coordination with local and central governments, running telephone help-lines offering advice and emotional support for isolated people:

We switched from a Monday to Friday, nine to five op-eration, to a seven day a week operation. …We were even doing things like contacting very local suppli-ers, and shops, to see if we could get deliveries put in place for people. (Participant 3, voluntary and com-munity sector)

### Predicted mechanisms of change and outcome

The predicted mechanism of change was that people would isolate themselves, with the outcome of avoiding infection. Some respondents expressed anxiety about the loss of liberty this represented, even though '*it was really quite an honourable aim*' (Participant 1, policy maker). Initially, shielding people were advised to avoid all contacts with others, even within the home. Respondents were aware that this was problematical:

Everybody realised that wasn't realistic for ninety per cent of people, who don't live in houses with west wings and east wings. And so… it then be-came a household isolation, which was even worse. (Participant 1, policy maker)

Equally, some of those shielding would require care at home, with care workers a potential source of infection:

A number of these people were vulnerable, they had comorbidities, they required support to manage that, and therefore contact would occur. (Participant 5, policy maker)

Respondents were aware that adherence to shielding advice would vary between individuals, and also over time, but felt that overall risk would still be reduced:

It's not an all or nothing, it's not if you break it once, you've broken the law, like with the legislative regula-tions. (Participant 2, policy maker)

### Intended impact

The primary intended impact was a reduction in mortality among CEV people. This was emphasised strongly by most respondents.

The idea is that if we isolate them they're less likely to get it, because if they get it they're probably going to be really, really poorly and die. (Participant 5, policy maker)

Some respondents described a broader range of bene-ficial impacts, including limiting the burden on the NHS, engendering of community spirit and more social and voluntary support for isolated vulnerable people.

It's about protecting the NHS, because that's in all our interests, isn't it, but actually on a local level, it was very much about supporting our communities. (Participant 10, local government)

### Risks/unintended impact

All participants described potential risks or unintended impact of shielding, and in particular the impact which it might have on mental health, through isolation and anxiety:

Some people who are shielding are still able to be effectively engaged somehow with society. They're working from home or whatever. But others have suffered, probably a lot…It created a whole lot of knock-on anxiety and everybody who was related in any way with the shielded person [was] put in a state of heightened awareness and heightened anxiety. (Participant 1, policy maker)

Concerns about the long-term mental health impact were reported as a factor in the decision to pause shielding advice in July 2020:

The effect on mental health started to outweigh the benefit once the prevalence was low enough. (Participant 2, policy maker)

In addition, concerns were widely expressed about the impact of shielding on physical health, including muscle wastage:

The debilitation from not leaving the house …this was not just that you were telling ninety- year-olds and eighty-year-olds, you were telling this to often quite fit young people who just had another condition to stay in the house. (Participant 4, policy maker)

Other unintended impacts included that on the work-force, as CEV people whose jobs could not be done from home were no longer available to work.

### Contextual factors

The shielding policy was based on the assumption that the CEV group would have enhanced protection over that of the general population. Almost as soon as shielding started, a lockdown was introduced for the wider popula-tion, imposing legally mandated restrictions on spending time outside the home except for certain exemptions. The national lockdown also brought certain whole-population initiatives, such as the 'furlough scheme' to subsidise wages for staff on temporary leave.

Respondents reflected that although lockdown was likely to have slowed the spread of the virus in the wider population, continuing steady rates of infection meant that shielding needed to continue to run in parallel:

It went on an awful lot longer than was envisaged in the first instance. (Participant 4, policy maker)

Since the shielding policy was developed in the context of a crisis, it—along with the associated shielding

programme—was devised and implemented at an unusually fast rate, and participants spoke of a clear shared purpose:

> We all had to make decisions differently and quickly, and we all had to use a bit more common sense than the traditional waiting for somebody else to come up with a strategy. (Participant 8, local government)

Local government respondents reported that there was a blurring around the edges of the population receiving their input, as they added people they knew to be vulnerable to their lists for this type of support, in addition to CEV people on the shielding list.

> Our proactive calling quickly extended to older people who were not necessarily shielding, but we felt they were also vulnerable – a number of them were… in effect shielding themselves. (Participant 10, local government)

In addition to the formal social support available from local authorities and third sector organisations, many CEV people were likely to have been able to access a wide range of other resources within the community, which enabled shielding to happen, especially informal support from family and friends. However, some would have lacked informal support from family, friends and neighbours and may have become increasingly isolated.

## DISCUSSION
### Principal findings
Our logic model provides a visual presentation of our understanding of the programme theory underlying the shielding intervention introduced across the UK to protect the most vulnerable people doing the early phases of the COVID-19 pandemic. It captures the key components of the shielding intervention (policy plus support programme), and identifies the mechanism by which it might make a difference, potential impacts, both unintended and intended, and key contextual factors. The logic model will underpin our evaluation of the impact of the shielding intervention, and we will continue to review it as the evaluation is finalised, in order to provide a structure to evaluating an intervention which to an extent was fluid and extemporised.

Our interviews with the key stakeholders add three main areas of insight: into the iterative and fluid process of developing and implementing shielding; into the range of activities which local government and the third sector introduced to support shielding people, outside the formal bounds of the programme; and into the feelings of many of those involved in the process, who revealed uncertainty about the logic and the justice of shielding.

The shielding policy, like many other aspects of the UK government's response to SARS-CoV-2, was introduced at pace. This left those involved in implementing the policy and delivering the shielding programme to work out the details at speed once the decision had been made. The policy implementation and support programme were designed and iteratively refined by the many different parties collaborating on the work across the four nations of the UK—SAGE, civil servants and clinicians, and those involved in providing services at local and community levels. In the delivery of the shielding support programme, there was some blurring, as the formal shielding policy ended up being delivered alongside support interventions for those regarded as vulnerable for social rather than clinical reasons. This does not undermine the programme theory, but instead is integral to it.

The nature of implementation as a process over time could also be observed. Although the basic principle of shielding remained the same throughout the time period of the intervention, the details evolved significantly—in terms of the nature of the advice, the definition of who should be on the shielding list and, in turn, the numbers of those included. As the pandemic continued, the evolution of the innovative attempts to deal with its impact was obvious, with new ways of working emerging between national government, local government and the VCS.

There was some ambiguity about who was doing the shielding: the use of the term 'Shielded Patient List' in guidance from the Department of Health and Social Care[15] implies that those at risk were being shielded by the state; yet the advisory nature of the guidance suggests that people were being asked to actively shield themselves—shielding rather than being shielded.

### Context of other literature
The changes in the intervention over time noted in our study reflect the evolving nature of the shielding list, and the slippage between guidance and advice in public discourse has been tracked in detail by Herrick.[16] Emerging evidence suggests that, despite the shielding intervention, there were still high rates of infection, hospitalisation and mortality in the shielding group[3 17] casting doubt on the mechanism proposed. This may in part be due to the impracticality of truly isolating people, particularly those who were in contact with clinical care providers and carers due to their vulnerability. Modelling has suggested that an 'imperfect' but realistic shielding strategy, in which contacts for those shielding were reduced by 80%, would still allow high rates of infection of high-risk individuals, with deaths estimated at 150–300% higher than under an implausible 'perfect' shielding model in which contacts were reduced to zero.[18] High rates of nosocomial COVID-19 infection have been identified likely to disproportionately affect CEV people.[19–21]

Our findings confirm previous studies which have identified the crucial role of local government and VCS organisations in supporting the implementation and operationalisation of the shielding policy,[22] and previous work discussing local variation in how the CEV list was created and the proportion and make-up of population on it,[1] in particular associated with the addition of people to the list by local clinicians.[23] Communication with CEV people to inform them of the support available has been

reported elsewhere as missing thousands of people, as records were incomplete.[23] A rapid evaluation in Scotland concluded that while the principle of shielding was valid, the intervention should not be repeated in exactly the same format.[24]

The concerns expressed in our interviews about potential negative impacts of shielding reflect suggestions from other studies that people advised to shield may have experienced increased anxiety and mental ill health and struggled to access routine healthcare,[1 25] and there may have been an additional strain on unpaid carers who were left without their usual support.[26]

### Strengths and limitations

This is the first study to develop a logic model examining all components of the UK COVID-19 shielding policy and programme, and to report the perspective of those involved in operationalising it.

In this phase of the EVITE Immunity evaluation, we did not record the perspective of CEV people themselves, who will have adhered to a greater or lesser extent, with adherence perhaps changing over time[11]; later phases of EVITE Immunity will explore the experience of this group.

### Implications

The UK's response to the pandemic of advising shielding for the most vulnerable was an unusual one, paralleled most closely in Europe by Ireland and Sweden, where a policy of shielding people aged over 70 contrasted with the general population lockdowns of its Nordic neighbours.[27] Although initial modelling in the UK explored a similar community level, age-based approach to shielding,[28 29] the adoption instead of an approach based on identifying and targeting individual clinical vulnerability was substantially more complex, and perhaps was regarded as more acceptable to the public. The emphasis in the UK on a mitigation approach—reducing the peak of infection while protecting the vulnerable—was soon overtaken by the imposition of general population lockdowns, but the two policy approaches continued to run in parallel, at least initially. Modelling early in the pandemic had identified a potential trade-off between increasing protection of the vulnerable and relaxing restrictions on the non-vulnerable.[11] However, the overlapping lockdown and shielding restrictions in the UK, together with the 'leakage' of shielding through necessary personal contacts, along with rates of full compliance reported as down to less than two-thirds of those on the list by early summer 2020,[30] make it hard to measure how this has played out. The impact of shielding will have depended in part on people's willingness to comply with guidance, shaped in turn by media narrative and social norms. There was no equivalent guidance on second-level shielding, for those essential close contacts of the shielded population, and measures to control infection within the wider population were imperfect.

Over time, there has been an evolution of risk within the shielded population resulting from subsequent waves of coronavirus infection, along with the vaccination programme. Shielding has now formally ended in both England and Wales. The most recent government guidance to previously shielding people in England reassured that, with protection from the vaccine, they would no longer be at substantially greater risk than the rest of the population,[31] though new mutations challenge the efficacy of existing vaccines. A now much better defined core high-risk subset remains, consisting of those unable to respond to vaccination appropriately.[32]

In terms of both infection rates and mortality in the UK, the pandemic has been identified as having an unequal impact on the population, reflecting health inequalities ultimately rooted in social inequalities.[32] The shielding policy may have exacerbated these, as those living in more crowded accommodation would have found it more challenging to maintain isolation from others.

## CONCLUSION

The shielding intervention was introduced to save lives by protecting the most vulnerable to SARS-CoV-2 infection. The shielding programme of support was introduced particularly rapidly and involved novel collaborations between various agencies. Components varied slightly but were broadly similar across the UK. It was a hitherto largely untested strategy based on 'common sense' risk mitigation rather than evidence-based interventions.

Naturally, this large-scale initiative created challenges both for those attempting to implement the policy and for those meant to benefit from it. Our logic model allows us to understand the different impacts (intended and unintended) of the shielding programme on organisations and populations, and spells out its rationale, components and mechanisms. Developing the logic model with input from key stakeholders has given additional insight to help us understand the causal links which will inform our evaluation of the impact of shielding and help us to understand its effect and limitations.

**Author affiliations**
[1]Swansea University Medical School, Swansea University, Swansea, UK
[2]School of Medicine, Cardiff University, Cardiff, UK
[3]Warwick Medical School, University of Warwick, Coventry, UK
[4]Public Contributor, SUPER group, Swansea, UK
[5]Delivery Unit, NHS Wales, Cardiff, UK
[6]College of Human and Health Sciences, Swansea University, Swansea, UK
[7]Administrative Data Research Unit, Welsh Government, Cardiff, UK

**Contributors** AP contributed to the conception and design of the study, to data acquisition, analysis and interpretation, and led the drafting of the paper. BE and VAW contributed to the conception and design of the study, to data acquisition, analysis and interpretation, and critically reviewed the draft paper. LD and LG contributed to the conception and design of the study, contributed to data analysis and interpretation and critically reviewed the draft paper. AA, AC-S, JD, AE, AJ, SJ, MRK, RL, JM, BS and AW contributed to the conception and design of the study and critically reviewed the draft paper. HS led the study, contributed to the conception and design of the study and critically reviewed the draft paper. AP is the guarantor

and accepts full responsibility for the work and/or the conduct of the study, had access to the data, and controlled the decision to publish.

**Funding** This work is supported by the National Core Studies Immunity Programme (NCSi4P).

**Competing interests** RL, SJ, AJ and AE are members of the Welsh Government COVID-19 Technical Advisory Group (TAG). AJ is also cochair of the Scientific Pandemic Insights Group on Behaviours (SPI-B) which is a subgroup of the Scientific Advisory Group for Emergencies (SAGE) advising the UK government. SJ is also a member of the Welsh Government Testing TAG and Cardiff University COVID Strategic Advisory Board (SAB).

**Patient and public involvement** Patients and/or the public were involved in the design, or conduct, or reporting, or dissemination plans of this research. Refer to the Methods section for further details.

**Patient consent for publication** Not applicable.

**Ethics approval** The EVITE Immunity study has received approval from the Newcastle North Tyneside 2 Research Ethics Committee (IRAS 295050).

**Provenance and peer review** Not commissioned; externally peer reviewed.

**Data availability statement** No data are available.

**ORCID iDs**
Alison Porter http://orcid.org/0000-0002-3408-7007
Ashley Akbari http://orcid.org/0000-0003-0814-0801
Jeremy Dale http://orcid.org/0000-0001-9256-3553
Adrian Edwards http://orcid.org/0000-0002-6228-4446
Ann John http://orcid.org/0000-0002-5657-6995
Ronan Lyons http://orcid.org/0000-0001-5225-000X

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
