## [Reviewer comments · BMJ Open]

ARTICLE DETAILS

TITLE (PROVISIONAL)	Rationale for the shielding policy for clinically vulnerable people in the UK during the COVID-19 pandemic: a qualitative study
AUTHORS	Porter, Alison; Akbari, Ashley; Carson-Stevens, A; Dale, Jeremy; Dixon, Lucy; Edwards, Adrian; Griffiths, Lesley; John, Ann; Jolles, Stephen; Kingston, Mark; Lyons, Ronan; Morgan, Jennifer; Sewell, Bernadette; Williams, Victoria; Snooks, Helen

VERSION 1 – REVIEW

REVIEWER	Peixoto, VASCO RICOCA FREIRE DUARTE Universidade Nova de Lisboa, Nova National School of Public Health
REVIEW RETURNED	29-Mar-2023

GENERAL COMMENTS	The authors have adressed most of my previous questions Still, it would be great to improve discussion in the text and abstract arround : 1. compliance and social norms, and media narratives by family and care workers2. The idea of shielding is undermined by lack of minimum general control measures such as masks in public transports.3. An idea that close contacts of vulnerable people should also be allowed to shield themselves in other contexts - 2nd level shielding and should be motivated to do so. They are a relevant target group for communication and training if shielding is to have 4.4. Shielding without isolations is possible but requeires discipline in control measures , (p2 masks, ventilation, distance, hanf hygine) and should be provided to care workers and family memebers to avoid isolation and negative health consequences of isolation in health.
--

VERSION 1 – AUTHOR RESPONSE

The reviewer made a number of related points about the contextual limitations on shielding, in terms of changes to behaviour in the wider population and compliance among those asked to shield. I have expanded on discussion of these issues in the text by making additions to the abstract and the implications sections as shown below.	
Still, it would be great to improve discussion in the text and abstract arround :	Added to the abstract:

1. compliance and social norms, and media narratives by family and care workers
2. The idea of shielding is undermined by lack of minimum general control measures such as masks in public transports.
3. An idea that close contacts of vulnerable people should also be allowed to shield themselves in other contexts - 2nd level shielding and should be motivated to do so. They are a relevant target group for communication and training if shielding is to have
4. Shielding without isolations is possible but requires discipline in control measures , (p2 masks, ventilation, distance, hand hygiene) and should be provided to care workers and family members to avoid isolation and negative health consequences of isolation in health.

Conclusions

Shielding was a largely untested strategy, aiming to mitigate risk by placing a responsibility on individuals to protect themselves.

Added to 'implications' section of the discussion, p14: 'The impact of shielding will have depended in part on people's willingness to comply with guidance, shaped in turn by media narrative and social norms. There was no equivalent guidance on second level shielding, for those essential close contacts of the shielded population, and measures to control infection within the wider population were imperfect.'